# Clinical Applications of Bee Venom Acupoint Injection

**DOI:** 10.3390/toxins12100618

**Published:** 2020-09-27

**Authors:** Ting-Yen Lin, Ching-Liang Hsieh

**Affiliations:** 1Department of Chinese Medicine, China Medical University Hospital, Taichung 40447, Taiwan; u103030067@cmu.edu.tw; 2Graduate Institute of Acupuncture Science, College of Chinese Medicine, China Medical University, Taichung 40402, Taiwan; 3Chinese Medicine Research Center, China Medical University, Taichung 40402, Taiwan

**Keywords:** bee venom, acupoint injections, clinical application, pharmacological mechanism

## Abstract

Bee venom is a complex natural mixture with various pharmaceutical properties. Among these properties, its peptides and enzymes have potential medical therapy for pain relief and inflammation. In clinical settings, this therapy has been used widely to treat diseases by injecting into acupoints. In this article, we have conducted various research from PubMed, Cochrane Library, and Clinical Key from inception of July 2020. The results revealed that bee venom therapy has been reported effective in anti-inflammatory, antiapoptosis, and analgesic effects. Moreover, bee venom acupuncture has been commonly used for clinical disorders such as Parkinson disease, neuropathic pain, Alzheimer disease, intervertebral disc disease, spinal cord injury, musculoskeletal pain, arthritis, multiple sclerosis, skin disease and cancer.

## 1. Introduction

Bee venom (BV) is a complex natural mixture produced by *Apis mellifera*, an European honey bee [1,2]. It is composed of different peptides, proteins, and bioactive components, including melittin, phospholipase A2 (PLA2), and apamin [3]. These components have various pharmaceutical properties, including anti-inflammatory, antinociceptive [4,5], antiapoptosis, antiarthritic [6], and anticancer effects [1,2,7].

BV therapy has been used for a long time [1] in various forms including the administration of live bee stings, the injection of BV, and BV acupuncture (BVA) [8]. Despite its clinical complications, toxicity or anaphylaxis, it is a safe form of acupuncture [6,9,10,11] and it has interesting pharmacological properties from a biotechnological point of view. Due to the bioactive compounds isolated from BV and its pharmacological actions, BVA is an effective treatment that is commonly used. Numerous studies of BVA have emerged in the last 20 years [12,13], and it has gradually become a popular alternative treatment [14,15,16]. BVA has mostly been used to treat intervertebral disc disease [14,17,18], neuropathic pain [19], arthritis [20,21], and skin diseases [22,23].

In this review, we will be summarizing current knowledge and numerous previous findings of BVA. We will also focus on its possible molecular mechanisms and clinical applications with respect to neural system diseases [24], musculoskeletal disease [25,26], autoimmune disease [2,8], skin disease [22,23], and cancer.

## 2. Pharmacological Mechanisms of the Most Abundant Components of BV

### 2.1. Melittin

Melittin is one of the main elements from A. mellifera venom [3], making up 40%–60% of its completely dry weight, it takes the major biological part in BV. This linear peptide contains 26 amino acids (NH2-GIGAVLKVLTTGLPALISWIKRKRQQ-CONH2) [27]. At low doses, melittin has anti-inflammatory, antibacterial, antifungal, antiarthritic, and antinociceptive effects, cytotoxicity against cancer cells, and activates PLA2 [1]. The potential anti-inflammatory mechanism (Figure 1) may function by decreasing the phosphorylation of IkB kinase (IKK), an inhibitor of nuclear factor kappa B (IκB) and nuclear factor kappa B (NF-κB) [28,29]. IKK not only reduces the secretion of tumor necrosis factor-α (TNF-α) [24,30], but also decreases the expressions of phospho-p38 [30], interleukin-1β (IL-1β), and interleukin-6 (IL-6) [27]. With the inhibition of extracellular regulated protein kinases/p38 mitogen-activated protein (ERK/p38 MAP) kinases, phosphorylation, and NF-κB pathways, matrix metalloproteinase-9 (MMP-9) may also be inhibited in expression and activity [31]. The potential apoptosis pathway causes the elevations in [Ca^2+^]_i_ and activation of PLA2 [32,33] by opening Ca^2+^ channels. In addition, melittin releases cytochrome c, increases levels of caspase-9, and increases the expressions of Bcl-2 associated X (Bax), attenuated B-cell lymphoma 2 (Bcl-2) [24], apoptosis-induced factors (AIFs), and endonuclease G (EndoG) [33]. 

### 2.2. PLA2 

BV therapy has been used for treating immunological diseases for many years. Recently, researchers have found that PLA2, which makes up 10%–12% of dry BV [3], has an immunomodulatory effect. Research has proven that regulatory T cells (Treg) can alleviate the development of autoimmune diseases [34,35] by promoting CD4^+^CD25^+^ Treg cell differentiation (Figure 2) [35]. PLA2 can bind to cluster of differentiation 206 (CD206), a dendritic cells (DC) surface receptor, and stimulate the secretion of prostaglandin E2 (PGE2). When PGE2 binds to prostaglandin E2 receptor 2 (EP2), forkhead box P3 (Foxp3) will increased expression, causing naïve T helper cells to differentiate into CD4^+^CD25^+^Foxp3^+^ Treg cells [35,36,37]. Moreover, research showed the portion of resting Treg cell (CD62LhiCD127low) has increased with bvPLA2 treatment [36]. Ye et al. [38] suggested that bvPLA2 has a neuroprotective effect through microglial deactivation as well as reduction in CD4^+^ T cell infiltration.

### 2.3. Apamin

Apamin is an 18-amino acid peptide and also the smallest neurotoxin in BV [2]. However, it makes up 37%–44% of dry BV and is essential to its biological function [3]. Alvarez-Fischer et al. [39] proved that apamin has anti-inflammatory and antinociceptive functions. Apamin, a blocker of Ca^2+^-activated K^+^ channels, influenced the permeability of cell membranes toward potassium ions [40]. Moreover, apamin is also able to transport across blood–brain barrier and affect the function of the central nervous system.

## 3. Clinical Applications and Mechanism of BV Injection

### 3.1. Neural System Diseases

#### 3.1.1. Parkinson Disease

##### Preclinical Studies

Parkinson disease is a typical neurodegenerative problem [41] caused by the selective loss of dopaminergic neurons of the substantia nigra pars compacta [42,43]. Its symptoms are characterized by resting tremors, rigidity, bradykinesia, and postural instability [44]. Recently, alternative therapies such as BV are increasing in number. From analyzing an acute 1-methyl-4-phenyl-1,2,3,6-tetrahydropyridine (MPTP)-induced mouse model, an animal model of Parkinson disease, Doo et al. [45] suggested that BVA could effectively protect dopaminergic neurons against MPTP toxicity, possibly through the inhibition of Jun activation [45] and the phosphoinositide 3-kinases/protein kinase B (PKB) pathway (also known as Akt [PI3K/Akt]) [46]. Another study investigated the anti-inflammatory effects of BV in Parkinson disease. After MPTP injection, it revealed a remarkable decrease in T helper cells (TH+). However, TH+ cells demonstrated a notable survival rate in animals when treated with BVA at ST36 [47]. The potential mechanism of BVA in Parkinson’s disease works by reinstating both brain TNF-α and IL-1β in brain tissues, inducing a significant elevation in caspase-3 activity and elevating the expression of apoptosis-related genes such as caspase-3 and Bax [46,48]. In another study, it examined the compounds that play the most effective roles in Parkinson disease and revealed that BV and apamin could protect against MPTP-induced dopaminergic cell loss by increasing striatal dopamine levels. Treatment with BV instead of apamin could increase TNF-α levels in MPTP-induced animal models [39]. Other studies have suggested that bvPLA2 may enhance the motor dysfunction and also pathological functions associated with Parkinson disease in human A53T α-Syn mutant transgenic (A53T Tg) mice by reducing the appearance of α-Syn, the activation of microglia, and the ratio of M1/M2 (M1 microglia produce toxic substances to neurons; M2 microglia generate anti-inflammatory as well as tissue repair elements to enhance survival and repairment) [49].

##### Clinical Studies

Furthermore, in terms of clinical trials, researchers of another study recruited 43 grownups with idiopathic Parkinson disease that were randomly separated into 3 groups: acupuncture, BVA, and without-treatment groups. The results revealed that the BVA and acupuncture groups both had significant improvements in the Unified Parkinson’s Disease Rating Scale, the Berg Balance Scale, and 30-m walking time [42,50]. Thus, BVA may play an supporting role in preventing the development of Parkinson’s disease [51].

#### 3.1.2. Neuropathic Pain

##### Preclinical Studies

BVA has another role in alleviating neuropsychiatric disorders [20]. Cold allodynia is an important symptom of neuropathic pain. Some research have studied whether treatment with BV injection could reduce cold allodynia in rats with chronic constriction injuries (CCIs) of the sciatic nerve. In one study, diluted BV (DBV) was injected (low-dose group: 0.25 mg/kg, high-dose group: 2.5 mg/kg) into the ST36 acupoint 2 weeks after sciatic nerve CCI for each group; and 2 weeks of low-dose DBV (0.25 mg/kg) injection for both groups. High dose of DBV (2.5 mg/kg) notably decreased cold allodynia since it can completely block an intrathecal pretreatment of idazoxan (α2-adrenoceptorantagonist) [52]. Other research has revealed that BVA attenuated pain through α₂-adrenergic receptors, instead of α₁-adrenergic receptors [52,53,54,55,56,57]. Combined with the inhibition of peripheral β-adrenoceptors, BVA could complement the strategies for decreasing inflammatory pain [58]. Other receptors, such as α4β2 nicotinic acetylcholine receptors, may also be influenced by BVA, and alleviates cold allodynia in oxaliplatin-injected rats [59]. Moreover, numerous articles have proved that BVA increases Fos expression and reduces nociceptive behavior [60,61,62]. In addition, intrathecal clonidine-induced analgesia was notably improved in efficacy when it combined with BVA [19].

#### 3.1.3. Peripheral Neuropathy

##### Preclinical Studies

Some chemotherapeutic agents induce peripheral neuropathy [63] because of the damage to sensory and motor nerve of the peripheral nervous system [64]. Oxaliplatin is an anticancer drug that leads to neuropathic cold allodynia [65,66,67,68]. Some articles have discussed whether BVA could decrease mechanical allodynia. According to the study by Yoon et al., after mice were injected with DBV (0.1 mg/kg) at ST36 of the right hind limb for 2 weeks and followed by oxaliplatin (10 mg/kg) injection, the mice had notably reduced ipsilateral mechanical allodynia. In the study, pre-injecting 2% lidocaine (40 mg/kg) into ST36, or yohimbine (25 µg/mouse), an alpha-2 adrenoceptor antagonist, into the spinal cord will both completely blocked the antiallodynic effect of DBV. However, the effect of DBV did not change with the pretreatment of naloxone (20 µg/mouse), an opioid receptor antagonist [66]. Similarly, another study revealed that the antiallodynic effect of BVA was related to the serotonin system and could be blocked by methysergide (mixed 5-HT1/5-HT2 receptor antagonist, 1 mg/kg i.p.) or MDL-72222 (5-HT3 receptor antagonist, 1 mg/kg i.p). That is, serotonin levels in the spinal cord could be elevated using BVA, and it was suggested that BVA alleviated cold allodynia in oxaliplatin-injected rats mainly through the activation of spinal 5-HT3 receptors [67,68]. Paclitaxel, a chemotherapy drug used against various tumors, also induced painful peripheral neuropathy. In this study, BV decreased the side effects of paclitaxel. In addition, the antihyperalgesic effect of BVA treatment at acupoint ST36 was more pronounced than at L11. Moreover, to investigate which of these compounds, BVA (1 mg/kg), melittin (0.5 mg/kg), and PLA2 (0.12 mg/kg), performed best analgesic effect when injected into ST36; the results demonstrated that melittin was the strongest analgesic among all [53].

##### Clinical Studies

Yoon et al. [64] report BV pharmacopuncture can reduce the scores of World Health Organization Common Toxicity Criteria for Peripheral neuropathy, Patient Neurotoxicity Questionnaire and Visual Analogue System, and also can improve the life quality in 11 Chemotherapy-induced peripheral neuropathy patients with weeks follow-up.

#### 3.1.4. Alzheimer Disease

##### Preclinical Studies

Alzheimer disease is a neuroinflammatory disease caused by the accumulation of 2 hallmark proteins, amyloid-β peptides and neurofibrillary tangles [69]. However, CD4+Foxp3+ Treg can modulate the neuroinflammation by suppressing the Th effector cell activation [38]. Treatment with bvPLA2 remarkably increased the numbers of Treg cells. Ye et al. suggested that bvPLA2 treatment may inhibit the progression of Alzheimer disease in 3xTg mice through increased glucose metabolism in the brain, decreased amyloid beta deposits in the hippocampus, reduced neuroinflammatory responses, and lower the percentage of CD4+ T cells infiltrating in the hippocampus [38]. These results indicate that BV may protect the brain tissue by reducing neuroinflammation.

#### 3.1.5. Intervertebral Disc Disease

##### Preclinical Studies

Intervertebral disc disease is a spinal disease that often causes neurological pain and dysfunction. Typically, anti-inflammatory and analgesic medicines are used to decrease nociceptive signals and ameliorate suffering. However, Tsai et al. revealed that BVA effectively alleviated pain symptoms [17]. Forty dogs with neurological disorder triggered by intervertebral disc disease underwent separate treatment for 6 weeks one treated by BV the usual therapy (oral prednisone and the nonsteroidal anti-inflammatory drug [NSAID] carprofen) another by usual therapy alone. BVA considerably improved grades recorded with the myelopathy scoring system as well as functional numeric scale [17].

#### 3.1.6. Spinal Cord Injury

##### Preclinical Studies

Spine injury is an inflammatory response that stimulated the secretion of proinflammatory cytokines, consisting of IL-1β and IL-6 [70,71,72]. Furthermore, spinal cord injuries lead to the increasing level of anti-inflammatory cytokines such as interleukin-10 (IL-10) [73]. One model of spinal cord compression showed that BVA at acupoints GV3 and ST36 notably increased the secretion of IL-10 in the first 6 h and reduced the expression of IL-6 in 24 h after treatment. However, there is no influence on the level of IL-1β and interleukin-4 (IL-4) cytokines. [74]. In another experiment on below-level neuropathic pain in rats with spinal cord injuries, DBV injection at acupoints can not only decreased glia expression and neuropathic pain but also enhanced the speed of recovery [75].

#### 3.1.7. Central Poststroke Pain

##### Clinical Studies

Following cerebrovascular incidents, central poststroke pain often manifests as sensory disturbances and neuropathic pain [76]. Pharmacopuncture is a widely used intervention for poststroke pain [77]. In a single-blind randomized controlled clinical trial, they recruited 20 patients who were suffering from central poststroke pain. These patients were injected with either DBV or saline at acupoints twice a week for 3 weeks. After treatment, the BV injection group showed more pronounced decreases in visual analog scale (VAS) scores than the saline injection group did [78]. A systematic review surveyed 138 potential articles, in which only four were randomized controlled trials that meeting the researchers’ criteria. The review demonstrated that injection with BV was more effective than injection with saline in decreasing poststroke shoulder pain [79].

### 3.2. Muscle and Skeletal Disease

#### 3.2.1. Musculoskeletal Pain

##### Clinical Studies

More and more people are affected by chronic neck pain or lower back pain. These patients are often prescribed NSAIDs [80] or alternative treatments such as BVA [81,82]. In one recent blinded human trial, 54 people suffering from chronic lower back pain were assigned to either BVA or sham acupuncture groups. Each patient received six treatments over 3 weeks. Participants in the BV group had a notable reduction in pain (26%) after 3 weeks, as recorded using the VAS [83,84]. In another study, 60 participants were randomly treated with either normal saline or BV injection. Both groups were treated twice a week for 4 weeks with injections at six acupoints (BL23, BL24, and BL25 on bilateral sides). The results revealed that BVA was more effective in treating chronic lower back pain.

BV injection at acupoints was more effective than normal saline aqua-acupuncture (used as a placebo) in the treatment of 30 patients suffering from acute ankle sprains after daily treatments for 1 week [25].

#### 3.2.2. Osteoarthritis-Related Knee Pain

##### Clinical Studies

Osteoarthritis is a type of arthritis that often affects the knee joints. BV can relieve the pain associated with knee osteoarthritis. BVA strongly stimulated aromatase activation in human leukemic cell line FLG19.1 and human osteoblast cells, enabling estrogen production by bone-derived cells to inhibit the development of osteoarthritis [85]. In one study, 60 patients with knee osteoarthritis received BVA twice a week continued for 4 weeks, and BVA was proven to be more effective than traditional acupuncture for pain relief [86]. In another study, 538 participants received weekly dermal injections of BV or histamine. After 12 weeks, the BV biotherapy group showed a remarkable improvement over the control group in Western Ontario and McMaster Universities Arthritis Index (WOMAC) pain scores [87]. Another comparative study revealed that a combination of intra-articular injection with intra-acupoint injection was more effective than conservative treatments for osteoarthritis of the knee [88]. 

### 3.3. Autoimmune Disease

#### 3.3.1. Rheumatoid Arthritis

##### Preclinical Studies

Rheumatoid arthritis is an autoimmune disorder characterized the proliferation of synovium and cellular infiltration that can lead to progressive joint destruction, deformity, disability, swelling, and pain in multiple joints [89,90]. Low-dose methotrexate is commonest treatment for rheumatoid arthritis. However, one side effect of methotrexate is hepatotoxicity, which can result in poor medication compliance [91]. Therefore, patients are likely to seek alternative treatments. Previous studies have indicated that BV injection may have anti-inflammatory and antinociceptive effects on rheumatoid arthritis in rats [92]. Li et al. [29] used Freund’s adjuvant-induced animal model to induced arthritis and found that treatment with BV at ST-36 notably suppressed Fos expression in the superficial layer of the lumbar spinal cord and thus reduced paw edema and nociceptive behaviors in the injected side of the paw. On the other hand, in a type-II collagen-induced arthritis (CIA) model, Lee et al. [93] found that BV injection therapy will inhibit immune responses. TNF-α production was notably lower in the BV group than in the control group, while IL-1β remained the same level [93]. Another study revealed that free radical levels and protease activity were significantly increased in rats with CIA compared with normal rats. BV injection (0.25 mg/kg) was an effective modulator of rheumatoid arthritis, inhibiting protease activities and removing reactive oxygen species (ROS) [94]. In addition, in combination use with methotrexate, BV may have increased bioavailability and antiarthritic effects [95]. Reduced hepatotoxicity was mostly due to the decreased expression of TNF-α and NF-kB (p65) in the synovial membrane of the hind paw [91]. These studies indicated that BV injection had antiarthritic and antinociceptive effects on rats with arthritis and can be an alternative therapy for rheumatoid arthritis [1].

#### 3.3.2. Multiple Sclerosis

##### Preclinical Studies

Multiple sclerosis is an autoimmune disease potentially leads to brain and spinal cord disabling diseases due to inflammation of the central nervous system. The etiology of multiple sclerosis is unknown. It may be induced by viral infection, environmental factors, or genetic factors [96]. Since 2000, several alternative medications have been used to treat multiple sclerosis, such as BV, cannabis, cannabinoids, acupuncture, and acupressure [97]. Some experiments have proven that BVA had effects on patients with multiple sclerosis. Pretreatment with BVA suppressed demyelination, glial activation, expression of cytokines such as interferon (IFN)-γ, interleukin-17 (IL-17), interleukin-17 A (IL-17A), TNF-α, IL-1β, and chemokines (monocyte chemotactic protein-1 (MCP-1) and macrophage inflammatory protein (MIP)-1α). In an animal model of multiple sclerosis, BVA will induce nitric oxide synthase (iNOS) activity, activate p38 mitogen-activated protein kinase (MAPK) and promote NF-κB signaling pathways in the spinal cords of rats with acute myelin basic protein (MBP)(68-82)–induced experimental autoimmune encephalomyelitis (EAE). BVA treatment may decrease the number of CD4^+^, CD4^+^/IFN-γ^+^, and CD4^+^/IL-17^+^ T cells and increase the number of CD4^+^/Foxp3^+^ T cells in the spinal cords and lymph nodes of rats with acute EAE [96].

### 3.4. Skin Disease

#### 3.4.1. Acne

##### Preclinical Studies

Acne is a chronic skin problem occurring in the sebaceous units of the skin, including on the face, neck, back, and chest [22]. *Propionibacterium acnes* is a key factor leading to the inflammatory reaction of acne. Traditionally, antibiotics are the first-line treatment. Nonetheless, antibiotics may result in the emergence of antibiotic resistant pathogens and side effects. Recent research has focused on the anti-inflammatory effects of BV. Heat-killed *P. acnes* increased levels of TNF-α, interleukin-8 (IL-8) and IFN-γ in HaCaT and THP-1 cells. BV inhibited the Toll-like receptors (TLR2) expression in HaCaT cells [28,98], thus suppressing the secretion of proinflammatory cytokines TNF-α and IL-1β [23,99]. Another study indicated that TLR activation happens during an inflammatory response, and will resulted in the activation of MAPK and transcription factor NF-κB signaling pathways. *P. acnes* increased the expression of IKK, IκB, and NF-κB in HaCaT cells, and lead to inflammation. Meanwhile, melittin will reduced the phosphorylation of IKK, IκB, and NF-κB [28,98].

#### 3.4.2. Atopic Dermatitis

##### Preclinical Studies

Atopic dermatitis is an inflammatory skin disorder caused by a deficient skin barrier, pruritus, eczema, dry skin, and an abnormal immunoglobulin E (IgE)–mediated allergic response to diverse external antigens [23]. In a mouse model of trimellitic anhydride (TMA)–induced skin impairment, BV (0.3 mg/kg) delivered at the BL40 acupoint reduced levels of both T helper cell type 1 (Th1) and Th2 cytokines and inhibited the synthesis of IL-4 and IgE. That is, BVA relieved inflammatory and allergic responses in the TMA model [100].

### 3.5. Cancer

#### Preclinical Studies

Cancer is the second leading cause of death worldwide. More and more new agents, treatment strategies, and biotoxins such as BV are being used to fight against cancer cells. One article investigated whether BV was efficient as an antitumor agent. Moon et al. indicated that through BV injection, Bcl-2, extracellular regulated protein kinases (ERK), and protein kinase B (Akt), will decrease in expression, and lead to apoptosis of cancer cells [101]. BV stimulated the expression of death receptors and inhibition of NF-κB, leading to the apoptosis of lung cancer cells [102]. BV also inhibited Akt and MAPK phosphorylation in Lewis lung carcinoma cells and human umbilical vein endothelial cells. And the expression of VEGF and VEGFR-2 were down-regulated [103]. In addition, BV intervened in calcium fluctuation, ROS generation, and the release of AIFs and EndoG in human melanoma cells [104]. In MDA-MB-231 breast cancer, Raman spectroscopy with multivariate analysis proved the effect of nuclear fragmentation and protein degradation at different concentrations [105]. In summary, with appropriate usage, BV may become a promising antineoplastic drug. 

## 4. Safety

People concern about the safety and side effects of BVA, since this treatment may lead to sensitization and even fatal anaphylactic reactions. In some cases, after the BV treatment, patients developed acute anaphylactic shock, Guillaume-Barre syndrome or irreversible ulnar nerve injury [20]. Ahn et al. [106] reported administered BVA (0.2 ml) became more redness and itching after few minutes. However, purified essential BV (0.2 ml) presented less allergic than original BV. Both of these pharmacopuncture showed similar anti-inflammatory effects [106]. A study of dose toxicity test in 30 Sprague-Dawley rats treated repeatedly with SBV (0.28 mg/kg) or normal saline (0.2 mL/kg) for 4 weeks showed no significant adverse effect for BV injection after 13 weeks [11]. Comparably, a 13-week repeated research suggested that the dose of BV under approximately 0.07 mg/kg is relatively safe without observed adverse effect [10].

## 5. Conclusions

BV injection at acupoints has been reported to be effective for clinical applications including treatment for inflammation, pain, arthritis, Parkinson disease, and cancer, among others. The effectiveness of BV injection at acupoints possibly results from its anti-inflammatory, antinociceptive, and antiapoptosis effects. However, more randomized, double-blinded, controlled clinical trials are required to confirm the scientific evidence of its therapeutic efficacy, and more pharmacological studies should be conducted to understand its therapeutic mechanisms. In the future, we expect BV injection at acupoints to become a new strategy for treating certain diseases.

## 6. Materials and Methods

We searched databases consisting of PubMed, Clinical Key, and the Cochrane Library from their inception to May 2020 making use of the following headings and keywords alone or in varied combinations: “Bee Venom (BV)”, “Bee Venom Injections”, “Acupoints”, and “Acupuncture”. The results were limited to articles in English only. We included supplementary articles after screening a manual review of the reference lists of reliable studies and review articles. We excluded articles where the full text was not available, articles unrelated to the therapeutics of BV injections at acupoints, articles with limited information, articles duplicated between databases, and case reports. A flowchart of the search procedures is presented in Figure A1, and a summary of the main published applications of BV therapy is shown in Table 1.

## Figures and Tables

**Figure 1 toxins-12-00618-f001:**
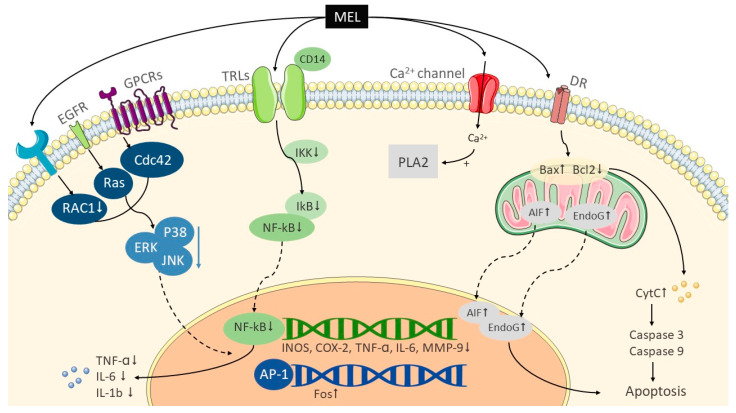
Possible mechanisms of melittin. Solid lines show signaling paths as well as the proteins associated with them; dotted lines show that the kinases or protein complexes were sent into the nucleus; down arrows show paths inhibiting or deactivating substrates or gene expression; horizontal arrows indicate pathways activating substrates or gene expression. AIFs, apoptosis-induced factors; AP-1, activator protein 1; Bax, Bcl-2 associated X; Bcl2, B-cell lymphoma 2; CD14, cluster of differentiation 14; Cdc42, cell division cycle protein 42; COX-2, cyclooxygenase-2; Cyt C, cytochrome c; DR, death receptor; EGFR, epidermal growth factor receptor; EndoG, endonuclease G; ERKs, extracellular regulated protein kinases; GPCRs, G protein-coupled receptors; IκB, inhibitor of nuclear factor kappa B; IKK, IkB kinase; IL-1β, interleukin-1β; IL-6, interleukin-6; INOS, inducible nitric oxide synthase; JNK, c-Jun N-terminal kinase; MEL, melittin; MMP-9, matrix metalloproteinase-9; NF-kB, nuclear factor kappa B; P38, p38 mitogen-activated protein kinases; RAC1, ras-related C3 botulinum toxin substrate 1; TLRs, toll-like receptors; TNF-α, tumor necrosis factor-α.

**Figure 2 toxins-12-00618-f002:**
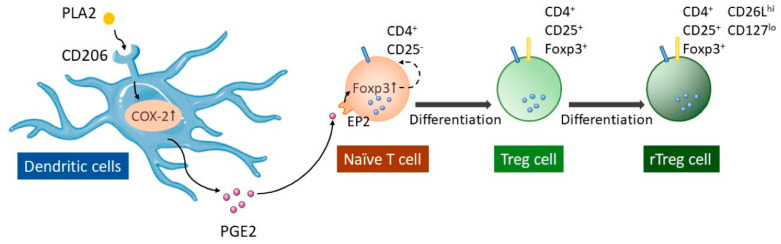
The possible mechanisms of (phospholipase A2) PLA2. Solid line arrows indicate signaling pathways; dotted line arrows indicate that Foxp3 increases the expression, resulting in naïve T helper cells differentiating into Tregs. CD206, cluster of differentiation 206; COX-2, cyclooxygenase-2; EP2, prostaglandin E2 receptor 2; Foxp3, forkhead box P3; PGE2, prostaglandin E2; PLA2, phospholipase A2; rTreg cell, resting regulatory T cell; Treg, regulatory T cell.

**Table 1 toxins-12-00618-t001:** Summary references of bee venom injection at acupoints related to clinical application.

Species	Size (N)	Venom/Compound	Acupoints	Dose	References/Results
Parkinson’s disease					
C57BL/6 mice	18	BV	Bilateral GB34	0.02 mL	Doo et al., 2010 [45] Prevented loss of tyrosine hydroxylase immunoreactivity and attenuated phospho-Jun immunoreactivity.
C57BL/6 mice	48	BV	Bilateral ST36	0.6 mg/kg	Kim et al., 2011 [47] BVA will decrease expression of the inflammation markers MAC-1 and iNOS in the SNpc.
C57BL/6 mice		BV/apamin	i.p. injections	Bee venom (low = 12 mg/kg; high = 120 mg/kg) Apamin (low: 0.5 mg/kg/BW; high 1.0 mg/kg/BW)	Alvarez-Fischer et al., 2013 [39] Apamin can partially reproduced the protective effects of the dopaminergic neurons.
Swiss albino mice	40	BV	GB34 bilaterally	0.02 ml	Khalil et al., 2015 [48] BVA lower the caspase-3 activation and apoptosis genes (Bax, Bcl2) expression in comparison to l-dopa in brain of rotenone treated mice.
A53T α-Syn transgenic mice		bvPLA2	intraperitoneal treatment	0.2 or 1 mg/kg	Ye et al., 2016 [49] After bvPLA2 injection, it was shown inhibit motor impairment, α-syn pathology, enhances microglial deactivation in the spinal cord and normalizes the ratio of M1/M2 microglial phenotypes.
Human	43	BV	Bilateral GB 20, LI 11, GB 34, ST36, and LR3	0.1 mL	Cho et al., 2012 [42] Improvement on the UPDRS, the Berg Balance Scale, and the 30 m walking time.
Human	11	BV	Bilateral GB20, LI4, GB34, ST36, and LR3	0.1 mL	Doo et al., 2015 [51] Combined treatment with BVA and acupuncture showed remarkable progress in gait speed, PDQL score, UPDRS scores.
Human	63	BV	Bilateral GB20, LI11, GB34, ST36, and LR3	0.1 mL	Cho et al., 2018 [44] A significant improvement was observed in UPDRS part II + III, part II, and part III scores, PIGD score.
Neuropathic Pain					
ICR mice	63	BV	CV12	0.25 mg/kg	Kwon et al., 2001 [55] BVA can produce antinociception via alpha 2-adrenoceptors, but not naloxone-sensitive opioid receptors.
Sprague-Dawley rats	28/33/18	BV	ST36	0.25 mg/kg	Roh et al., 2004 [54] BVA remarkably reduces the thermal hyperalgesia and this antihyperalgesic effect will activate alpha2-adrenoceptors.
Sprague-Dawley rats	18	BV	ST36	0.8 mg/kg	Kwon et al., 2004 [62] BV acupoint stimulation activates brainstem catecholaminergic neurons.
ICR mice	53	BV	ST36	High: 10 mg/kg Middle: 0.1 mg/kg Low: 0.001 mg/kg	Roh et al., 2006 [61] Low dose of BV increase in Fos immunoreactive neurons in the ipsilateral; middle dose only in the SDH and NECK areas of the ipsilateral spinal dorsal horn; high dose throughout much of the ipsilateral dorsal horn consisting of the SDH, NP, and NECK areas.
ICR mice/Sprague-Dawley rats	15	BV	ST36	Mice:0.25 mg/kg/20 µL Rats:0.01 mg/kg/50 µL	Yoon et al., 2009 [19] BVA produced a significant clonidine-induced analgesia.
ICR mice	8/groups	BV	ST36	0.8 or 0.08 mg/kg	Kang et al., 2011 [56] BVA triggered activation of spinal astrocytes and this inhibition is associated with spinal alpha-2 adrenoceptors.
ICR mice		BV	ST36 or control points (SP9 or GB39 or tail base)	1 mg/mL	Kim et al., 2011 [60] BVA administrated at ST36 may active the central α₂-adrenergic as well as the peripheral nerve and modulate METH-induced hyperthermia, hyperactivity and Fos expression.
Sprague-Dawley rats	24/16	BV	ST36	0.25 or 2.5 mg/kg	Kang et al., 2012 [52] DBV could active spinal α2-adrenoceptor and alleviate CCI-induced cold allodynia.
ICR mice		BV	ST36	0.8 mg/kg	Kang et al., 2013 [58] Antinociceptive impact of BVA can be improved by inflection of adrenal medulla-derived epinephrine and this impact is moderated by peripheral β-adrenoceptors.
Sprague-Dawley rats	27	BV	GV3	0.25 mg/kg	Yoon et al., 2015 [59] Spinal α4β2 receptors, nicotinic acetylcholine receptors, but not muscarinic receptors, moderate the suppressive impact of BVA.
Peripheral Neuropathy					
C57BL/6 mice	17/18	BV	Right ST36	0.1 mg/kg	Yoon et al., 2013 [66] BVA reduces ipsilateral mechanical allodynia depending on spinal cord alpha-2 adrenoceptors.
Sprague-Dawley rats	24	BV	LI11, ST36, GV3	1.0 mg/kg	Lim et al., 2013 [65] BVA reduces oxaliplatin-induced cold allodynia via alpha-2 adrenoceptors.
Sprague–Dawley rats	25/34	BV	GV3	0.25 mg/kg	Lee et al., 2014 [68] BVA reduces oxaliplatin-induced acute cold allodynia through the activation of serotonergic system and spinal 5-HT3 receptors.
C57BL/6 mice	59	BV	Right ST36	0.1 mg/kg	Yeo et al., 2016 [107] BVA decreased oxaliplatin-induced mechanical allodynia and recovered the loss of IENFs through an α-2 adrenoceptor mechanism.
C57BL/6 mice	25	BV	Right ST36	0.25, 1, and 2.5 mg/kg	Kim et al., 2016 [67] The combine treatment of BVA and morphine is moderated by spinal opioidergic and 5-HT3 receptors which could decrease oxaliplatin-induced neuropathic pain.
Sprague-Dawley rats	14/11/28/36	BV/melittin/phospholipase A2	ST36, LI11	BVA (1 mg/kg) melittin (0.5 mg/kg) phospholipase A2 (0.12 mg/kg)	Choi et al., 2017 [53] BVA could decrease paclitaxel-induced neuropathic pain with spinal α2-adrenergic receptor.
Human	11	melittin	Bilateral GB39, LV3: lower extremities neuropathy. Bilateral LI4, SJ5, GB39, and LV3: patients with both upper and lower extremities neuropathy.	0.1 mL	Yoon et al., 2012 [64] After BVA, both PNQ scores and WHO CIPN grade decreased. VAS has also decrease.
Alzheimer Disease					
3xTg AD mice	27	bvPLA2	Intraperitoneal injection	0.2 mg/kg 1 mg/kg	Ye et al., 2016 [38] PLA2 has neuroprotective effect via reduction in CD4^+^ T cell infiltration and microglial deactivation.
3xTg-AD mice	50	bvPLA2	Intraperitoneal injection	0.5 mg/kg	Baek et al., 2018 [69] BvPLA2 could trigger the amelioration AD pathology and Tregs
Intervertebral Disk Disease					
canines	40	BV	Bilateral LI 04, SI 03, KI 03, ST 36, BL 23, BL 40, GB 30, GB 34, and LR 03, unilateral GV01, Baihui, and Ashipoints	0.1 mL (20 μg)	Tsai et al., 2015 [17] BV injection exerted a strong effect on canines with moderate to severe IVDD and reduced clinical rehabilitation time.
Spinal Cord Injury					
Wistar rats	3-4 animals/group	BV	BVA: GV3 and ST36; nonacupoints AT: no treatment	20 µL diluted in saline (0.08 mg/kg)	Raquel Nascimento de Souza et al., 2017 [74] BVA increased the expression of IL-10 at 6 h and reduced the expression of IL-6 at 24 h after SCI compared with the controls.
Sprague-Dawley rats	16	BV	ST36	0.25 mg/kg(50-µL)	Kang et al., 2015 [75] BVA assisted in motor function recuperation as suggested by the Basso-Beattie-Bresnahan score.
Central Post-Stroke Pain					
Human	20	BV	LI15, GB21, LI11, GB31, ST36 and GB39 of the affected side	0.05 ml	Cho et al., 2013 [78] After BVA, there is significant decreases in visual analogue pain scores.
Musculoskeletal Pain					
Human (low back pain)	54	BV	BL23, BL24, BL25, GB30, GV3, GV4, GV5	0.2 mL for the first week, 0.4 mL for the second week, and 0.8 mL for the third week	Seo et al., 2017 [83] After BV injection, Beck’s Depression Inventory and Oswestry Disability Index, EuroQol 5-Dimensionshowed improved.
Osteoarthritis Knee Pain					
Human	60	BV	SP10, ST34, ST36, GB34, LR3, Ex-LE2, Ex-LE5	0.1 mL	Y-B Kwon et al., 2001 [86] After BV injection computerized infrared thermography (IRT) and pain relief scores showed significant improved.
Human	69	BV	ST35, GB34, EX32, ST36, SP9, Ashipoints	0.1 mL	Lee et al., 2012 [88] BV injection exhibited significant improvement on VAS and KWOMAC effects when treating knee OA.
Human	358	BV	knee top, eye-1 medial, eye-2 lateral, ST 34, BL40, BL5, BL19, BL21, BL23, BL25, and BL27	100 μg	Conrad et al., 2019 [87] HBV biotherapy resulted in significant improvements in VAS of knee OA pain and physical function.
Rheumatoid Arthritis					
Sprague- Dawley rats	60	BV	ST36	1 mg/kg	Kwon et al., 2001 [90] BV treatment inhibit paw edema also decreased arthritis-induced nociceptive behaviors.
Sprague-Dawley rats	60	BV	ST36	0.9 mg/kg	Kwon et al., 2002 [92] BVA suppressed the increase of IL-6 caused by RA and decreased arthritis-induced nociceptive behaviors.
Sprague-Dawley rats	90	whole BV	Intraperitoneal injection	one bee/rat	Kang et al., 2018 [21] BVA showed erosions in inflammatory cell infiltrations and articular cartilage.
DBA/1 mice	27	BV	ST36	0.1 mL	Lee et al., 2004 [93] BV reduced the progression of arthritis and lead to the inhibition of the immune responses.
Sprague–Dawley rats	80	BV	ST36	0.25 mg/kg	Baek et al., 2006 [89] BVA can alleviate inflammatory pain via alpha2-adrenergic receptor.
Lewis rats	12	BV	bilateral Shinsu (B23)	50 µL/kg	Suh et al., 2006 [94] BVA is an effective RA modulator, preventing protease activities and eliminating ROS.
Sprague–Dawley rats	12	BV	proximal tibialis anterior muscle around the right knee	0.8 mg/kg	Yang et al., 2010 [108] After BVA there is more improved on the weight load test and revealed lower activity in bone scintigraphy.
Sprague-Dawley rats	88	BV/melittin	ST36	BV (1 mg/kg/day) melittin (0.5 mg/kg/day)	Li et al., 2010 [29] BV and melittin therapies statistically decreasedarthritis-induced nociceptive behaviors.
Wister rats	80	BV	ST36	0.5 mg/kg	Darwish et al., 2013 [91] BVA ameliorated TNF-α and the over expression of NF-κB in liver induced by methotrexate.
Wistar rats	47/39	BV	subcutaneously	0.25 mg	Yamasaki et al., 2015 [95] Treatment with MTX or BV alone will ameliorate edema. MTX is more effective in reducing hyperalgesia than BV. However, anti-arthritic effect of BV is better than MTX.
Multiple sclerosis					
Lewis rats/C57BL/6 mice		BV	ST36 placebo acupoints: SP9, GB39, NA1, NA2, NA3, NA4	0.25 and 0.8 mg/kg	Lee et al., 2016 [96] BVA with ST36 could attenuate the progression of EAE by increasing T cells and suppressing T-helper 1and T-helper 17 responses.
Skin disease					
BALB/c mice	50	BV	BL40	0.3 mg/kg	Sur et al., 2016 [100] BVA inhibited the proliferation of T cells, the synthesis of Th1 and Th2 cytokines, and the production of immunoglobulin E and IL-4.
Cancer					
C57BL/6JmsSlc mice		BV	subcutaneously	0.01, 0.1 or 1 mg/kg	Huh et al., 2010 [103] BV prevented MAPK and AKT phosphorylation and down modulated activation of vascular endothelial growth factor, which can suppress the vascular endothelial growth factor-induced proliferation and the viability of Lewis lung carcinoma

5HT3, 5hydroxytryptamine receptor; α-syn, α-synuclein; AKT, protein kinase B; AT, classic or general acupuncture; Bax, Bcl-2 associated X; Bcl2, B-cell lymphoma; bvPLA2, bee venom phospholipase A2; BVA, bee venom acupuncture; CCI, chronic constriction injury; CIPN, chemotherapy-induced peripheral neuropathy; DBV, dilute bee venom; ICR mice, introduction of C57BL6 mouse; IENFs, intraepidermal nerve fibers; iNOS, inducible nitric oxide synthase; IL-10, interleukin-10; IL-6, interleukin-6; IVDD, intervertebral disc disease; SDH, superficial dorsal horn; M1, M1 microglia produce toxic substances to neurons; M2, M2 microglia produce anti-inflammatory and tissue repair factors to promote survival and repair; MAC-1, macrophage-1 antigen; MAPK, p38 mitogen-activated protein kinase; MTX, methotrexate; NA1, nonacupoint 1; NA2, nonspecific acupoints near ST36; NA3 and NA4, nonspecific acupoints away from ST36; NECK, the neck region; NF-κB, nuclear factor kappa B; NP, nucleus proprius; PDQL, Parkinson’s Disease Quality of Life Questionnaire; PIGD, postural instability gait difficulty; PLA2, phospholipase A2; PNQ, Patient Neurotoxicity Questionnaire; RA, rheumatoid arthritis; ROS, reactive oxygen species; SNpc, substantia nigra pars compacta; TNF-α, tumor necrosis factor-α; UPDRS, Unified Parkinson’s Disease Rating Scale; VAS, visual analog scale.

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
