# Peer review of "Clinical Applications of Bee Venom Acupoint Injection"

_toxins, 2020, doi:10.3390/toxins12100618_

Round 1
Reviewer 1 Report
Although there is an article in the Toxins magazine with similar information [Moreno et al, 2015_Three Valuable Peptides from Bee and Wasp Venoms for Therapeutic and Biotechnological Use: Melittin, Apamin and Mastoparan_Toxins 7 (4): 1126–1150], the review proposed by the authors brings more diverse and broader content about the different Clinical Applications and Pharmacological Mechanism of Bee Venom Acupoint Injection.
This reviewer believes that this review is very relevant to encourage researchers to develop biotechnological products and processes with potential for human and animal health.
This reviewer suggests that the authors indicate in the introduction of this article that, despite the clinical complications (toxicity and/or anaphylaxis) caused by multiple bee stings, bee venom and/or its isolated components have interesting pharmacological properties from a biotechnological point of view.
Bearing in mind that there are other minority components in bee venom, the ideal would be to highlight in the title of the topic "Potential Pharmacological Mechanisms of BV Components" that only the Pharmacological Mechanisms of the most abundant components of bee venom will be addressed in this topic (line 34).
Author Response
#[Reviewer 1]
- Although there is an article in the Toxins magazine with similar information [Moreno et al, 2015_Three Valuable Peptides from Bee and Wasp Venoms for Therapeutic and Biotechnological Use: Melittin, Apamin and Mastoparan Toxins 7 (4): 1126–1150], the review proposed by the authors brings more diverse and broader content about the different Clinical Applications and Pharmacological Mechanism of Bee Venom Acupoint Injection. This reviewer believes that this review is very relevant to encourage researchers to develop biotechnological products and processes with potential for human and animal health.
#RESPONSE:
- Thank you very much for the reviewer’s comments.
- This reviewer suggests that the authors indicate in the introduction of this article that, despite the clinical complications (toxicity and/or anaphylaxis) caused by multiple bee stings, bee venom and/or its isolated components have interesting pharmacological properties from a biotechnological point of view.
#RESPONSE:
- Thank you very much for the reviewer’s comments.
- Introduction: “BV therapy has been used for thousands of years [1] in various forms including the administration of live bee stings, the injection of BV, and BV acupuncture (BVA) [8]. BVA is a safe form of acupuncture [6, 9-11] in which BV is applied to acupoints to treat diseases.” had been revised into “BV therapy has been used for a long time [1] in various forms including the administration of live bee stings, the injection of BV, and BV acupuncture (BVA) [8]. Despite its clinical complications, toxicity or anaphylaxis, it is a safe form of acupuncture [6, 9-11] and it has interesting pharmacological properties from a biotechnological point of view.” (Line 59-63)
- Bearing in mind that there are other minority components in bee venom, the ideal would be to highlight in the title of the topic "Potential Pharmacological Mechanisms of BV Components" that only the Pharmacological Mechanisms of the most abundant components of bee venom will be addressed in this topic (line 34).
#RESPONSE:
- Thank you very much for the reviewer’s comments.
- “Potential Pharmacological Mechanisms of BV Components” had been revised into “Pharmacological Mechanisms of the Most Abundant Components of BV”. (Line 73)
Reviewer 2 Report
Major revision
The authors should include and describe the safety and potential side effects of bee venom acupuncture, since this kind of intervention could lead to important sensitization and even fatal anaphylatic reactions.
Minor revisions
- Figure 2: Change the DNA molecule by a nucleus image representation
- line 101: define Thelper?
- line 108: define influence - increase, descrease?
- line 112: toxic substances
- line 160: change to Th effector cells
- lines 177 and 180: change b to beta
- line 212: exlcude the expression that is
- line 217: We are interested in determining whether acupoint injections or local injections were more effective for treatment with BV. confuse period in the context - should be revised
- line 224: change and to comma
- line 229: Freund’s adjuvant-induced arthritis ??? animal model
- line 236: the dose should be expressed as micrograms per animal body weight
- line 248: yoga is not a medication
- line 256: change but to and
- line 262: change to key factor
- line 264: change to antibiotic resistent pathogens
- line 266: change to: inhibited the TLR-2 expression in HaCaT cells.....
- line280: Wrong information! - ischaemic heart disease is the leading cause of death, according WHO
- line 404: correct the citation
Author Response
# [Reviewer 2]
- The authors should include and describe the safety and potential side effects of bee venom acupuncture, since this kind of intervention could lead to important sensitization and even fatal anaphylatic reactions.
#RESPONSE:
- Thank you very much for the reviewer’s comments.
- As the reviewer’s comments.
- Safety: “People concern about the safety and side effects of BVA, since this treatment may lead to sensitization and even fatal anaphylactic reactions. In some cases, after the BV treatment, patients developed acute anaphylactic shock, Guillaume-Barre syndrome or irreversible ulnar nerve injury [20]. Ahn et al. [106] reported administered BVA (0.2ml) became more redness and itching after few minutes. However, purified essential BV (0.2ml) presented less allergic than original BV. Both of these pharmacopuncture showed similar anti-inflammatory effects [106]. A study of dose toxicity test in 30 Sprague-Dawley rats treated repeatedly with SBV (0.28 mg/kg) or normal saline (0.2 mL/kg) for 4 weeks showed no significant adverse effect for BV injection after 13 weeks [11]. Comparably, a 13-week repeated research suggested that the dose of BV under approximately 0.07 mg/kg is relatively safe without observed adverse effect [10].” had been added. (Line 375-386)
- Figure 2: Change the DNA molecule by a nucleus image representation
#RESPONSE:
- Thank you very much for the reviewer’s comments.
(2) “Figure 2” had been revised into as PDF. (Line 118-119)
- line 101: define T helper?
#RESPONSE:
- Thank you very much for the reviewer’s comments.
- “TH+ cells” had been revised into “T helper cells (TH)”. (Line 145-146)
- line 108: define influence - increase, decrease ?
#RESPONSE:
- Thank you very much for the reviewer’s comments.
- “influence” had been revised into “increase”. (Line 153)
- line 112: toxic substances
#RESPONSE:
- Thank you very much for the reviewer’s comments.
- “substances toxic” had been revised into “toxic substances”. (Line 157)
- line 160: change to Th effector cells
#RESPONSE:
- Thank you very much for the reviewer’s comments.
- “Th cell” had been revised into “Th effector cells”. (Line 223)
- lines 177 and 180: change b to beta
#RESPONSE:
- Thank you very much for the reviewer’s comments.
- “IL-1b” had been revised into “IL-1β”. (Line 244, 249)
- line 212: exlcude the expression that is
#RESPONSE:
- Thank you very much for the reviewer’s comments.
- “BVA strongly stimulated aromatase activation in human leukemic cell line FLG19.1 and human osteoblast cells, enabling estrogen production by bone-derived cells. That is, BVA inhibited the development of osteoarthritis [85].” had been revised into “BVA strongly stimulated aromatase activation in human leukemic cell line FLG19.1 and human osteoblast cells, enabling estrogen production by bone-derived cells to inhibit the development of osteoarthritis [85].”. (Line 284-286)
- line 217: We are interested in determining whether acupoint injections or local injections were more effective for treatment with BV. confuse period in the context - should be revised
#RESPONSE:
- Thank you very much for the reviewer’s comments.
- “We are interested in determining whether acupoint injections or local injections were more effective for treatment with BV.” had been deleted.
- line 224: change and to comma
#RESPONSE:
- Thank you very much for the reviewer’s comments.
- “Rheumatoid arthritis is an autoimmune disorder characterized by cellular infiltration and the proliferation of synovium that can lead to progressive joint destruction, deformity, disability, and swelling and pain in multiple joints [89, 90].” had been revised into “Rheumatoid arthritis is an autoimmune disorder characterized the proliferation of synovium and cellular infiltration that can lead to progressive joint destruction, deformity, disability, swelling, and pain in multiple joints [89, 90].”. (Line 298-300)
- line 229: Freund’s adjuvant-induced arthritis ??? animal model
#RESPONSE:
- Thank you very much for the reviewer’s comments.
- “Using Freund’s adjuvant-induced animal model, a study by Li et al. [29] found that treatment with BV at ST-36 notably suppressed Fos expression in the superficial layer of the lumbar spinal cord and thus reduced paw edema and nociceptive behaviors in the injected side of the paw” had been revised into “Li et al. [29] used Freund’s adjuvant-induced animal model to induced arthritis and found that treatment with BV at ST-36 notably suppressed Fos expression in the superficial layer of the lumbar spinal cord and thus reduced paw edema and nociceptive behaviors in the injected side of the paw”. (Line 304-308)
- line 236: the dose should be expressed as micrograms per animal body weight
#RESPONSE:
- Thank you very much for the reviewer’s comments.
- “5 µL/100 g” had been revised into “0.25 mg/kg”. (Line 312)
- line 248: yoga is not a medication
#RESPONSE:
- Thank you very much for the reviewer’s comments.
- “Since 2000, several alternative medications have been used to treat multiple sclerosis, such as BV, cannabis, cannabinoids, acupuncture, acupressure, and yoga” had been revised into “Since 2000, several alternative medications have been used to treat multiple sclerosis, such as BV, cannabis, cannabinoids, acupuncture, and acupressure [97].”. (Line 324-326)
- line 256: change but to and
#RESPONSE:
- Thank you very much for the reviewer’s comments.
- “BVA treatment may decrease the number of CD4+, CD4+/IFN-γ+, and CD4+/IL-17+ T cells but increase the number of CD4+/Foxp3+ T cells in the spinal cords and lymph nodes of rats with acute EAE” had been revised into “BVA treatment may decrease the number of CD4+, CD4+/IFN-γ+, and CD4+/IL-17+ T cells and increase the number of CD4+/Foxp3+ T cells in the spinal cords and lymph nodes of rats with acute EAE [96]”. (Line 333-335)
- line 262: change to key factor
#RESPONSE:
- Thank you very much for the reviewer’s comments.
- “Propionibacterium acnes is a factor” had been revised into “Propionibacterium acnes is a key factor”. (Line 340)
- line 264: change to antibiotic resistent pathogens
#RESPONSE:
- Thank you very much for the reviewer’s comments.
- “However, antibiotics may lead to the emergence of antibiotic pathogens and side effects.” had been revised into “Nonetheless, antibiotics may result in the emergence of antibiotic resistant pathogens and side effects”. (Line 341-343)
- line 266: change to: inhibited the TLR-2 expression in HaCaT cells.....
#RESPONSE:
- Thank you very much for the reviewer’s comments.
- “BV inhibited the expression of heat-killed P. acnes inhibited the Toll-like receptors (TLR2) in HaCaT cells”had been revised into “BV inhibited the Toll-like receptors (TLR2) expression in HaCaT cells”. (Line 344-346)
- line280: Wrong information! - ischaemic heart disease is the leading cause of death, according WHO
#RESPONSE:
- Thank you very much for the reviewer’s comments.
- “Cancer is still the leading cause of death worldwide.” had been revised into “Cancer is the second leading cause of death worldwide.”. (Line 362)
- line 404: correct the citation
#RESPONSE:
- Thank you very much for the reviewer’s comments.
- “!!! INVALID CITATION !!! [7].” had been revised into “Reference 31; Jeong, Y. J.; Cho, H. J.; Whang, K.; Lee, I. S.; Park, K. K.; Choe, J. Y.; Han, S. M.; Kim, C. H.; Chang, H. W.; Moon, S. K.; Kim, W. J.; Choi, Y. H.; Chang, Y. C., Melittin has an inhibitory effect on TNF-α-induced migration of human aortic smooth muscle cells by blocking the MMP-9 expression. Food Chem Toxicol 2012, 50 (11), 3996-4002.”. (Line 502-505)

Reviewer 3 Report
Comments:
- The article should focus only on clinical applications as it is titled. The tile is not 100% right in terms of its contents. The article's title can be modified, or the summary should focus on the clinical applications and limitations of the available clinical studies. The preclinical studies data can connect the clinical observations, and then the title can be the same. It will depend on how the authors present the review.
- The authors should separate the results from preclinical studies from the clinical studies. I found, in most cases, the information obtained from preclinical studies. The number of clinical studies is recorded less than preclinical studies, and it is expected.
- The list of abbreviations can be added as separated block after abstract as these are so many and often hard to remember.
- It is a narrative review. Figure 3 and relevant information can be deleted from the main manuscript and can be added as a supplementary document.
- In the main text, some information may be rewritten to make it more specific. For example: Line 94 of page 3. “Because of substantial limitations of conventional therapy….”. The limitations were never mentioned in the article.
- “3. Clinical Applications of BV Injection:” Some sections have no clinical data, but these are presented under the main caption of clinical applications. For example: “3.1.5. Intervertebral Disc Disease”, “3.1.2. Neuropathic Pain” There is no separation between the experimental and clinical studies. It will be straightforward to understand if the authors put the preclinical and clinical studies separately under each caption.
- The limitations and adverse events are not recorded in the manuscripts and not adequately discussed. It should be included in the article to understand the overall situation of bee venom treatment.
Author Response
# [Reviewer 3]
- The article should focus only on clinical applications as it is titled. The tile is not 100% right in terms of its contents. The article's title can be modified, or the summary should focus on the clinical applications and limitations of the available clinical studies. The preclinical studies data can connect the clinical observations, and then the title can be the same. It will depend on how the authors present the review.
#RESPONSE:
- Thank you very much for the reviewer’s comments.
- As the reviewer’s commonts.
- Title: “Clinical Applications and Pharmacological Mechanism of Bee Venom Acupoint Injection” had been revised into “Clinical Applications of Bee Venom Acupoint Injection”. (Line 2-3)
- The authors should separate the results from preclinical studies from the clinical studies. I found, in most cases, the information obtained from preclinical studies. The number of clinical studies is recorded less than preclinical studies, and it is expected.
#RESPONSE:
- Thank you very much for the reviewer’s comments.
- As the reviewer’s comments.
- “Clinical Applications of BV Injection” had been revised into “Clinical Applications and mechanism of BV Injection”. (Line 132)
- The subsection of “Preclinical studies” and “Clinical studies” had been added at each caption. (Line 135, 160, 169, 187, 211, 220, 231, 242, 254, 268, 282, 297, 320, 338, 353, 361)
- Peripheral neuropathy section, “Yoon et al. [64] report BV pharmacopuncture can reduce the scores of World Health Organization Common Toxicity Criteria for Peripheral neuropathy, Patient Neurotoxicity Questionnaire and Visual Analogue System, and also can improve the life quality in 11 Chemotherapy-induced peripheral neuropathy patients with weeks follow-up.” had been added (Line 211-218)
- The list of abbreviations can be added as separated block after abstract as these are so many and often hard to remember.
#RESPONSE:
- Thank you very much for the reviewer’s comments.
- The list of abbreviations had been added as separated block after abstract. (Line 19-51)
- Abbreviation: 5HT3, 5hydroxytryptamine receptor; A53T Tg, A53T α-Syn mutant transgenic mice; AIFs, apoptosis-induced factors; AKT, protein kinase B; AP-1, activator protein 1; AT, classic or general acupuncture; Bax, Bcl-2 associated X; Bcl2, B-cell lymphoma 2; BV, Bee venom; BVA, bee venom acupuncture; bvPLA2, bee venom phospholipase A2; CCI, chronic constriction injury; CD14, cluster of differentiation 14; CD206, cluster of differentiation 206; Cdc42, cell division cycle protein 42; CIA, collagen-induced arthritis; CIPN, chemotherapy-induced peripheral neuropathy; COX-2, cyclooxygenase-2; Cyt C, cytochrome c; DBV, dilute bee venom; DC, dendritic cells; DR, death receptor; EAE, experimental autoimmune encephalomyelitis; EGFR, epidermal growth factor receptor; EndoG, endonuclease G; EP2, prostaglandin E2 receptor 2; ERK, extracellular regulated protein kinases; ERK/p38 MAP, extracellular regulated protein kinases/p38 mitogen-activated protein kinases; ERKs, extracellular regulated protein kinases; Foxp3, forkhead box P3 ; GPCRs, G protein-coupled receptors; ICR mice, introduction of C57BL6 mouse; IENFs, intraepidermal nerve fibers; IFN, interferon; IgE, immunoglobulin E; IKK, IkB kinase ; IL-10, interleukin-10; IL-17 , interleukin-17; IL-17A, interleukin-17 A; IL-1β, interleukin-1β; IL-4, interleukin-4; IL-6, interleukin-6; IL-8, interleukin-8; iNOS, induced nitric oxide synthase ; iNOS, inducible nitric oxide synthase; IVDD, intervertebral disc disease; IκB, inhibitor of nuclear factor kappa B; IκB, inhibitor of nuclear factor kappa B; JNK, c-Jun N-terminal kinase; M1, M1 microglia produce toxic substances to neurons; M2, M2 microglia produce anti-inflammatory and tissue repair factors to promote survival and repair; MAC-1, macrophage-1 antigen; MAPK, p38 mitogen-activated protein kinase; MBP, myelin basic protein; MCP-1, monocyte chemotactic protein-1; MEL, melittin; MIP, macrophage inflammatory protein; MMP-9, matrix metalloproteinase-9; MMP-9, matrix metalloproteinase-9; MPTP, 1-me+7:116thyl-4-phenyl-1,2,3,6-tetrahydropyridine ; MTX, methotrexate; NA1, nonacupoint 1; NA2, nonspecific acupoints near ST36; NA3 and NA4, nonspecific acupoints away from ST36; NECK, the neck region; NF-κB, nuclear factor kappa B; NP, nucleus proprius; P38, p38 mitogen-activated protein kinases; PDQL, Parkinson’s Disease Quality of Life Questionnaire; PGE2, prostaglandin E2; PGE2, prostaglandin E2; PIGD, postural instability gait difficulty; PKB, phosphoinositide 3-kinases/protein kinase B; PLA2, phospholipase A2; PNQ, Patient Neurotoxicity Questionnaire; RA, rheumatoid arthritis; RAC1, ras-related C3 botulinum toxin substrate 1; ROS, reactive oxygen species; SDH, superficial dorsal horn; SNpc, substantia nigra pars compacta; Th1, T helper cell type 1; TLR2, Toll-like receptors; TLRs, toll-like receptors; TMA, trimellitic anhydride; TNF-α, tumor necrosis factor-α; Treg, regulatory T cells; UPDRS, Unified Parkinson’s Disease Rating Scale; VAS, visual analog scale; WOMAC, Western Ontario and McMaster Universities Arthritis Index; α-syn, α-synuclein.
- It is a narrative review. Figure 3 and relevant information can be deleted from the main manuscript and can be added as a supplementary document.
#RESPONSE:
- Thank you very much for the reviewer’s comments.
- “A flowchart of the search process is presented in Figure 3” had been revised into “Figure A. flowchart of the search procedures”.
- The figure was deleted from the main manuscript and was added as a supplementary document. (Line 421-423)
- In the main text, some information may be rewritten to make it more specific. For example: Line 94 of page 3. “Because of substantial limitations of conventional therapy….”. The limitations were never mentioned in the article.
#RESPONSE:
- Thank you very much for the reviewer’s comments.
- “Because of the substantial limitations of conventional therapy, alternative therapies such as BV are increasing in number.” had been revised into “Recently, alternative therapies such as BV are increasing in number”. (Line 138-140)
- Clinical Applications of BV Injection:” Some sections have no clinical data, but these are presented under the main caption of clinical applications. For example: “3.1.5. Intervertebral Disc Disease”, “3.1.2. Neuropathic Pain” There is no separation between the experimental and clinical studies. It will be straightforward to understand if the authors put the preclinical and clinical studies separately under each caption.
#RESPONSE:
(1) Thank you very much for the reviewer’s comments.
(2) As the reviewer’s comments.
(3) “Clinical Applications of BV Injection” had been revised into “Clinical Applications and mechanism. of BV Injection”. (Line 132)
(4) The subsection of “Preclinical studies” and “Clinical studies” had been added at each caption. (Line 135, 160, 169, 187, 211, 220, 231, 242, 254, 268, 282, 297, 320, 338, 353, 361)
- Peripheral neuropathy section, “Yoon et al. [64] report BV pharmacopuncture can reduce the scores of World Health Organization Common Toxicity Criteria for Peripheral neuropathy, Patient Neurotoxicity Questionnaire and Visual Analogue System, and also can improve the life quality in 11 Chemotherapy-induced peripheral neuropathy patients with weeks follow-up.” had been added (Line 211-218)
- The limitations and adverse events are not recorded in the manuscripts and not adequately discussed. It should be included in the article to understand the overall situation of bee venom treatment.
#RESPONSE:
- Thank you very much for the reviewer’s comments.
- Safety: “People concern about the safety and side effects of BVA, since this treatment may lead to sensitization and even fatal anaphylactic reactions. In some cases, after the BV treatment, patients developed acute anaphylactic shock, Guillaume-Barre syndrome or irreversible ulnar nerve injury [20]. Ahn et al. [106] reported administered BVA (0.2ml) became more redness and itching after few minutes. However, purified essential BV (0.2ml) presented less allergic than original BV. Both of these pharmacopuncture showed similar anti-inflammatory effects [106]. A study of dose toxicity test in 30 Sprague-Dawley rats treated repeatedly with SBV (0.28 mg/kg) or normal saline (0.2 mL/kg) for 4 weeks showed no significant adverse effect for BV injection after 13 weeks [11]. Comparably, a 13-week repeated research suggested that the dose of BV under approximately 0.07 mg/kg is relatively safe without observed adverse effect [10].” had been added. (Lines 375-386)

Round 2
Reviewer 2 Report
The authors made all the changes I had suggested and added information about the safety of the procedure.
Reviewer 3 Report
Happy with the improvements of the article in the revised version.